# Reduction and Growth Inhibition of *Listeria monocytogenes* by Use of Anti-Listerial Nisin, P100 Phages and Buffered Dry Vinegar Fermentates in Standard and Sodium-Reduced Cold-Smoked Salmon

**DOI:** 10.3390/foods12244391

**Published:** 2023-12-06

**Authors:** Even Heir, Merete Rusås Jensen, Anette Wold Aasli, Ingunn Berget, Askild Lorentz Holck

**Affiliations:** Nofima AS—Norwegian Institute of Food, Fisheries and Aquaculture Research, P.O. Box 210, N-1431 Ås, Norway; merete.rusas.jensen@nofima.no (M.R.J.); anette.wold.asli@nofima.no (A.W.A.); ingunn.berget@nofima.no (I.B.); askild.holck@nofima.no (A.L.H.)

**Keywords:** cold-smoked salmon, Listeria, salt, sodium reduction, nisin, bacteriophage, fermentate, microbiota, food safety

## Abstract

Cold-smoked salmon are ready-to-eat products that may support the growth of pathogenic *Listeria monocytogenes* during their long shelf-life. Consumption of such contaminated products can cause fatal listeriosis infections. Another challenge and potential risk associated with CS salmon is their high levels of sodium salt. Excess dietary intake is associated with serious health complications. In the present study, anti-listerial bacteriocin (nisin), P100 bacteriophages (Phageguard L, PGL) and fermentates (Verdad N6, P-NDV) were evaluated as commercial bio-preservation strategies for increased control of *L. monocytogenes* in standard (with NaCl) and sodium-reduced (NaCl partially replaced with KCl) CS salmon. Treatments of CS salmon with nisin (1 ppm) and PGL (5 × 10^7^ pfu/cm^2^) separately yielded significant initial reductions in *L. monocytogenes* (up to 0.7 log) compared to untreated samples. Enhanced additive reductions were achieved through the combined treatments of nisin and PGL. Fermentates in the CS salmon inhibited the growth of Listeria but did not lead to its eradication. The lowest levels of *L. monocytogenes* during storage were observed in nisin- and PGL-treated CS salmon containing preservative fermentates and stored at 4 °C, while enhanced growth was observed during storage at an abusive temperature of 8 °C. Evaluation of industry-processed standard and sodium-replaced CS salmon confirmed significant effects with up to 1.7 log reductions in *L. monocytogenes* levels after 34 days of storage of PGL- and nisin-treated CS salmon-containing fermentates. No differences in total aerobic plate counts were observed between treated (PGL and nisin) or non-treated standard and sodium-reduced CS salmon at the end of storage. The microbiota was dominated by *Photobacterium,* but with a shift showing dominance of *Lactococcus* spp. and *Vagococcus* spp. in fermentate-containing samples. Similar and robust reductions in *L. monocytogenes* can be achieved in both standard and sodium-replaced CS salmon using the bio-preservation strategies of nisin, PGL and fermentates under various and relevant processing and storage conditions.

## 1. Introduction

*Listeria monocytogenes* is a bacterial pathogen that can cause serious and fatal listeriosis, ranked third among foodborne infections with the highest mortality rate [1]. *L. monocytogenes* is among the five pathogens responsible for the greatest burden of costs of illness and loss of quality-adjusted life years (QALYs, [2]). Particular risk products are ready-to-eat foods with extended shelf-life with the ability to support the growth of this pathogen [3].

Listeria species are ubiquitously present in natural environments including raw, fresh salmon [4,5,6,7,8] and are often found in smoked fish production environments including machines for salting, skinning and slicing [9,10,11]. *L. monocytogenes* in the food processing environment is considered the primary source of post-processing contamination during food manufacturing [12,13].

Cold-smoked (CS) salmon is produced without any critical control points, and products contaminated with *L. monocytogenes* may therefore lead to potentially serious health, economic and social outcomes, including fatal cases, outbreaks, recalls and reduced trust in the salmon processing industry [3,14,15,16]. Cold-smoked fish (mainly CS salmon) is reported to be the food item that most often harbors *L. monocytogenes* and also has the highest proportion of products exceeding the critical limit of 100 cfu/g [17]. Occasionally, cold-smoked salmon with *L. monocytogenes* exceeding levels of 10^5^–10^6^ cfu/g have been reported [17,18,19]. *L. monocytogenes* is therefore considered the largest microbial challenge for the salmon industry.

The relatively high sodium content of CS salmon represents a public health issue and thereby constitutes another challenge for processors. High dietary sodium intake leads to an increased risk of hypertension and related cardiovascular diseases [20]. WHO Member States have agreed to reduce the global population’s intake of salt by a relative 30% by 2025 [21]. Potassium has proved to be useful for the replacement of sodium in many products. It provides many similar functional and technological properties, being an essential nutrient, contributing to salty taste and being an osmotic preservative. Challenges with extensive replacement of Na^+^ with K^+^ are the relatively offensive acrid, metallic and bitter side tastes of K^+^. Replacement of 25–33% of NaCl with KCl in CS salmon did not cause any sensory changes [22,23]. No differences in the growth of *L. monocytogenes* in sodium-reduced and standard CS salmon were detected [24,25]. The production of CS salmon with reduced sodium content, while maintaining or improving the sensory and microbial quality and safety, poses key challenges for the CS salmon processing industry.

Hygienic processing is pivotal to lowering the risk of *L. monocytogenes* in ready-to-eat (RTE) products, but no hygienic actions can guarantee listeria-free CS salmon. Therefore, additional anti-listerial strategies are required. These may include bactericidal and bacteriostatic bio-based preservation strategies enabling the production of safe, high-quality CS salmon. The principle of using bacteriocins (e.g., nisin) or bacteriophages to kill pathogenic microorganisms in foods has been known for years (see reviews [26,27,28,29]). Commercialized nisin (E234) is the only bacteriocin approved for food preservation. It is widely used in dairy and meat products [30]. The commercial product Phageguard L (PGL) based on the listeria phage P100 has been evaluated for its safety and efficacy in application on RTE foods and RTE seafood [31,32], as well as for anti-listerial effects in many different foods, such as meat [33,34], poultry [35], cheese [36] and fresh cut fruits [37]. When applied to CS salmon, both nisin and PGL result in initial reductions, but surviving *L. monocytogenes* may show growth during storage [38,39,40]. Strategies that improve their efficacy, either alone or by combinatory treatments, leading to increased kill and/or increased growth inhibition in CS salmon, are therefore highly relevant. Previous combinatory treatments including nisin and phages on CS salmon have included protecting cultures (*Carnobacteria*) or lauric arginate [38,39,41,42,43]. Organic acid salts (e.g., lactate, diacetate) are widely used in the meat industry while acetate-based fermentates also show potential for growth inhibition of *L. monocytogenes* on CS salmon [38,39,44]. Enhanced control can be obtained by combining growth-inhibiting strategies with methods that kill *L. monocytogenes* like UV light [45] or various compounds [46,47]. Of particular interest are FDA-approved, Generally Recognized As Safe (GRAS) bio-preservatives including commercialized nisin [43] and P100 bacteriophages (PGL).

The objectives of the current study were to determine and evaluate the effects of the commercial, bio-based listeria control strategies using listericidal nisin, P100 phages (PGL) or a combination of these treatments on CS salmon during relevant processing, treatment and storage conditions. The anti-listerial treatments were tested under different conditions, including the addition of growth-inhibiting fermentates to CS salmon (Verdad N6, Provian NDV (P-NDV)), and sodium-reduced CS salmon. Final evaluations included the anti-listerial effects of standard and sodium-reduced CS salmon produced in an industrial CS salmon smokehouse. Effects of the mitigation strategies on other quality parameters including total viable bacterial counts and the microbiota composition were also determined.

## 2. Materials and Methods

### 2.1. Experimental Overview and Study Design

The study included three experiments termed Experiments 1, 2 and 3 using various types of CS salmon to determine anti-listerial effects after selected treatments and storage conditions (see Table 1 for an overview of variables and parameters). Common parameters in Experiments 1 through 3 were the treatments with nisin and Phageguard L (PGL) applied separately and combined in addition to non-treated controls. The anti-listerial effects of preservative salts (acetate-based fermentates) as an ingredient in CS salmon were also included in all three experiments. Experiment 1 was conducted with standard CS salmon (with 3% NaCl, i.e., no sodium salt reduction). Additional experimental factors were Verdad N6, two strain mixes used for contamination and two storage temperatures (4 °C and 8 °C). The salmon was moderately smoked. The two strain mixes were an 8-strain mix and a 10-strain mix, with the latter including two strains with reduced sensitivity to PGL compared to the 8-strain mix, where all strains were PGL-sensitive (Table 2). Experiment 2 was conducted with two types of CS salmon: standard and sodium-reduced using KCl as a partial replacer of NaCl salt. An additional factor was the fermentate Provian NDV (P-NDV). The CS salmon was mildly smoked, contaminated with the 10-strain mix and stored at 4 °C after treatments. For Experiment 3, three types of salmon were included: standard, sodium-reduced (using a commercial KCl-based salt replacer) and sodium-reduced produced with added fermentate P-NVD. The salmon was smoked using the in-house standard smoking protocol applied by the manufacturer. Contamination and storage temperature were as in Experiment 2. For Experiments 1 and 2, CS salmon was produced in a research pilot plant facility. For the final evaluation (Experiment 3), CS salmon was produced in an industrial commercial smokehouse.

### 2.2. Bacterial Strains and Culture Conditions

*L. monocytogenes* strains used in the experiments are given in Table 2. The 10 strains included 6 strains isolated from salmon and salmon processing facilities, 3 strains associated with human listeriosis outbreaks and 1 strain from cattle. The strains represented three serovars commonly associated with human listeriosis and various multiple-locus variable-number tandem repeat analysis (MLVA) and multi-locus sequence types (ST). The strains were stored at −80 °C in brain heart infusion (BHI) broth with 15% glycerol. For each experiment, strains were cultured on BHI agar at 37 °C for 24 h, after which single colonies were picked to inoculate 2 mL BHI broth before incubation at 37 °C for 24 h. This pre-culture was used for inoculation (1%) of each strain in individual tubes of 2 mL BHI broth. After incubation at 37 °C for 24 h, the bacterial cultures (approximately 1 × 10^9^ CFU/mL of each culture) were mixed and diluted to contain equal cell numbers of each strain in two separate strain mixes, an 8-strain mix and a 10-strain mix. The 8- and 10-strain mixes (Table 2) were stored at 4 °C for 20–24 h for cold adaptation. Dilutions to working solutions (5 × 10^6^ CFU/mL) were performed in 0.9% NaCl.

### 2.3. Antimicrobial Compounds for L. monocytogenes Growth Inhibition and Reduction

Nisin (Merck Life Science AS, Oslo, Norway) solutions in distilled water were made from the purchased 2.5% stock (equivalent to 25,000 ppm) to obtain the in-use solutions in the range of 200–400 ppm nisin. The nisin solutions were freshly made and kept refrigerated until use. Commercial PGL (Micreos Food Safety BV, Wageningen, The Netherlands) was diluted in 0.9% NaCl to appropriate in-use concentrations and kept refrigerated until use. The Verdad N6 powder (Verdad N6) was obtained from Corbion (Amsterdam, The Netherlands). It is a white distilled vinegar produced by fermentation. P-NDV was from Niacet Corp. (Niagara Falls, NY, USA) and is a clean label, no-sodium, neutralized, dry buffered vinegar based on fermentation. These acetate-based fermentates were added in the dry-salting procedure to fresh salmon fillets prior to cold-smoking as described below.

### 2.4. Production and Preparation of Cold-Smoked Salmon

For the production of CS salmon for Experiments 1 and 2 (see Table 1 for details), fresh salmon fillets with skin were packed on ice at a salmon slaughterhouse. Upon receipt two days after filleting, the tail part of the fillets was discarded. The remaining fillets were divided into two parts and individually weighed. The salmon fillets were dry-salted by adding 3% (*w*/*w*) salt (NaCl or a mix of 2.1% NaCl and 0.9% KCl) and 1% Verdad N6 (Experiment 1) or 0.9% P-NDV (Experiment 2) when appropriate. The salmon fillets were dry-salted in plastic bags to avoid spills and obtain controlled salt levels. The bags with salted fillets were sealed under a mild vacuum and stored at 4 °C for 64–68 h to obtain even salt distribution in the fillets prior to smoking. The salmon fillets were thereafter unpacked, placed horizontally on stainless steel meshes and cold-smoked in a programmable smoking cabinet (DOLESCHAL, process control unit SC2000; Inject Star Maschinenbau GmbH, Hagenbrunn bei Wien, Austria) using smoke generated from beech chipwood (Räuchergold KL 2/16; J. Rettenmaier & Söhne GmbH, Rosenberg, Germany). The fillets were exposed to moderate smoking (Experiment 1) with four cycles of smoking, 30 min × 4. Smoking was performed at 25 °C and included a drying step of 30 min with air circulation, followed by cycles of smoking/smoke circulation for a total of approximately 3.5 h. Mild smoking (30 min × 2) was applied to CS salmon used for Experiment 2. The CS salmon was vacuum-packaged and stored at 0 °C for approximately 64 h to allow time for the diffusion of smoke components in the salmon fillets. The CS salmon was stored frozen (−40 °C) under vacuum and thawed prior to the *L. monocytogenes* contamination experiments. For use in Experiment 3, CS salmon was produced in an industrial smokehouse. Fresh salmon fillets were dry-salted and exposed to the salt for 24 h to achieve approximately 3% (*w*/*w*) NaCl, or 3% (*w*/*w*) NuTek 78,300 (NuTek; NuTek Natural Ingredience, Omaha, NE, USA, containing approximately 70% NaCl and 30% KCl) or 3% NuTek and 0.9% (*w*/*w*) P-NDV. Also, 0.6% sucrose was added to the fillets according to the producer’s standard recipe for CS salmon production. The filets were thereafter rinsed in fresh water, dried to remove excess water and smoked according to the factory’s standard commercial procedure before they were vacuum-packed and shipped ice-stored to the lab for contamination experiments. The CS salmon was stored frozen prior to the contamination experiments. Characteristics of the produced CS salmon (added and measured levels of NaCl, KCl and acetate) were as reported [25].

### 2.5. Contamination of Salmon with L. monocytogenes and Antimicrobial Treatments

Smoked salmon fillets were thawed. Slices of approximately 5 g were each added 20 µL of the 8- or 10-strain (Table 2) *L. monocytogenes* cocktail (5 × 10^6^ CFU/mL) on the 10 cm^2^ surface and left at 4 °C for 30 min. The samples containing approximately 1 × 10^4^ cfu/cm^2^ were then treated with 1 ppm nisin, 5 × 10^7^ pfu/cm^2^ phages or a combination thereof in 50 µL on the contaminated surface before non-inoculated 5 g slices of salmon were placed on the *L. monocytogenes*-contaminated salmon surface to obtain 10 g samples. The concentration of nisin was calculated to obtain nisin levels of 1 ppm if evenly spread throughout the 10 g salmon sample. The contaminated salmon samples reflecting a contaminated and treated, sliced CS salmon sample were put in separate stomacher bags and thereafter vacuum-packed and stored at 4 °C or 8 °C for up to 34 days. All experiments with *L. monocytogenes* were performed in a Biosafety level 3 pilot processing plant. In Experiment 2, control samples without added *L. monocytogenes* were packed and stored under identical conditions to assess the indigenous background microbiota of the CS salmon.

### 2.6. Culture-Dependent and Independent Microbial Analyses

Bacterial counts of *L. monocytogenes* in CS salmon stored at 4 °C and 8 °C were determined at days 0, 1, 7, 12, 19 and 29 after contamination (if not otherwise indicated). The samples were prepared and diluted before being plated on Rapid L’mono agar as previously reported [25]. Total counts were determined by plating of samples not inoculated with *L. monocytogenes* on blood agar plates and aerobic incubation at 15 °C for 5 to 7 days. Microbiota profiling using high-throughput sequencing of bacterial 16S rRNA gene amplicons (MiSeq, Illumina, San Diego, CA, USA) was performed on selected samples of CS salmon stored for 29 days. Sample preparations were as previously described [44]. In short, DNA was extracted from thawed pellets obtained from centrifuged supernatants followed by PCR, purification and quantification prior to sequencing using a MiSeq protocol provided by Illumina. Sequence analyses were carried out using QIIME (Quantitative Insights into Microbial Ecology, version 1.9.1). Sequences originating from salmon DNA (i.e., non-target DNA; unassigned in the openref analysis) were removed by filtration. The level 6 (genus) table derived from QIIME was used for bar chart illustrations.

### 2.7. Statistical Analyses

Experiments 1 and 2 were analyzed using analysis of variance (ANOVA) with average (across parallels) growth of *L. monocytogenes* (log_10_) as the dependent variable. For Experiment 1, the independent variables were days of storage, preservative salt (with/without Verdad N6), *L. monocytogenes* mix (8-strain or 10-strain) and treatment. Data for the two temperatures were analyzed separately. For Experiment 2, the independent variables were days of storage, salmon type (standard and sodium-reduced), preservative salt (with/without P-NDV) and the treatments. In both cases, two-way interactions were included while neglecting higher-order interactions. Pairwise comparisons were applied for treatment effects across CS salmon types (with/without fermentates and for different salt combinations). Comparisons with the control were performed using pairwise comparisons with Tukey adjustments of *p*-values or by setting up defined contrasts. Comparisons were made across the whole time span and for selected days. In Experiment 3, separate analyses were carried out for each type of the three types of CS salmon since the factorial approach could not be pursued with this setup. All analyses were performed using R (R_Core_Team, 2021 #4637). A significance level of α = 0.05 was used, meaning that samples were considered statistically different for *p*-values < 0.05.

## 3. Results

### 3.1. Experiment 1: Effects of Nisin and Phages on L. monocytogenes in CS Salmon with and without Preservative Fermentate Verdad N6

Levels of *L. monocytogenes* were significantly reduced in contaminated CS salmon treated with nisin, PGL or both compared to non-treated samples (Figure 1). All treatments exhibited initial listericidal effects (day 1 after treatment) for both strain mixes (8-strain and 10-strain mix). The *L. monocytogenes* levels were maintained at a significantly lower level throughout the entire 29-day storage period at 4 °C after separate treatments of nisin and PGL (mean reductions of 0.5–1.0 log; *p* < 0.01; Table 3). Treatments with PGL and nisin together showed marked enhanced initial reductions at day 1 (mean reductions of 1.6 log; *p* < 0.001) which were generally maintained throughout storage at 4 °C (mean reductions of 1.7 log; *p* < 0.001). Growth of *L. monocytogenes* was more rapid and reached higher levels when salmon was stored at an abusive temperature (8 °C, Appendix A). Similar differences in *L. monocytogenes* levels of treated versus non-treated samples were still observed for samples stored at 4 °C (Table 3) and 8 °C (Appendix A).

The addition of Verdad N6 alone did not lead to the eradication of *L. monocytogenes*, but significant growth inhibition during storage was observed. The salmon with Verdad N6 had lower levels of *L. monocytogenes* (*p* < 0.01) after day 7 and throughout the storage period (both 4 °C and 8 °C) across all treatments. Up to 2.0 log and 1.9 log, lower levels (*p* ≤ 0.001) of *L. monocytogenes* were observed in samples containing Verdad N6 treated with nisin and PGL after 29 days of storage at 4 °C and 8 °C, respectively, compared with non-treated samples. In Verdad N6 samples treated with PGL alone or PGL and nisin in combination, *L. monocytogenes* levels remained below the initial contamination level throughout 29 days of storage at 4 °C (Figure 1). ANOVA showed that treatments and Verdad N6 explained the vast majority (≥85%) of the variability in the data (Appendix A).

Of note, the PGL treatment showed effective reductions in samples containing either the 8-strain or the 10-strain mix despite the latter having two strains with reduced sensitivity to PGL (according to an in vitro susceptibility plate assay). The overall reductions in *L. monocytogenes* at day 29 were, however, 0.3–0.4 log higher (*p* < 0.05) for the 8-mix than the 10-mix for PGL and the combined treatment at day 29 in salmon stored at 4 °C.

### 3.2. Experiment 2: Effects of Nisin and Phages on L. monocytogenes in Mildly Smoked CS Salmon with and without KCl-Based Sodium Reduction

There were small differences in the *L. monocytogenes* reductions obtained by the treatments between standard (3% NaCl) and sodium-reduced (2.1% NaCl + 0.9% KCl) CS salmon during 29 days of storage. All treatments (nisin, PGL or both) significantly (*p* < 0.05) contributed to the initial killing of *L. monocytogenes* in sodium-reduced CS salmon, whereas only the combined treatment showed significant *L. monocytogenes* reductions in standard CS salmon. Only CS salmon treated with combined nisin and PGL showed consistent and significantly lower listeria levels than untreated samples after 29 days of storage (Figure 2; Table 3). Of note, the growth of *L. monocytogenes* was slightly more extensive and reached higher maximum levels (approximately 7 log cfu/cm^2^) compared with corresponding samples in Experiment 1. There were slightly higher overall *L. monocytogenes* levels (0.3 log, *p* < 0.01) in treated standard CS salmon vs. sodium-reduced samples, but this difference was not evident in salmon containing P-NDV. Salmon with P-NDV showed reduced *L. monocytogenes* growth and 1.6 log lower levels (mean reductions of all sampling days and treatments; *p* < 0.001) compared with levels in treated samples without the P-NDV fermentate in the entire 29-day storage period. ANOVA showed that treatments and P-NDV explained the vast majority (84%) of the variability in the data (Appendix A).

### 3.3. Experiment 3: Effects of Nisin and Phages on L. monocytogenes in Industrially Produced Standard and Sodium-Reduced CS Salmon Produced with and without Preservative Fermentate P-NDV

A final evaluation of the anti-listerial effects of listeria phages and nisin was performed on standard and sodium-reduced CS salmon produced in an industrial salmon smokehouse (Figure 3; Table 4). The CS salmon was processed according to standard processing conditions in the factory (see Section 2.4 for details). *L. monocytogenes* levels increased by 1.2 log during storage in untreated standard salmon while no significant growth of *L. monocytogenes* was observed in the sodium-reduced samples with or without added P-NDV (Figure 3). The overall anti-listerial effects of the nisin and PGL treatments were highly similar in the three types of salmon (Figure 3). The treatments showed initial reductions in listeria levels by 0.5–0.6 log (*p* ≤ 0.001) when PGL or nisin was used separately, while marked additive effects (1.4–1.7 log reductions, *p* ≤ 0.001) were evident with the double treatment. The significantly lower listeria levels in treated CS salmon were generally maintained throughout the 34-day storage period. Of note, in all treated samples, the significant initial reductions along with limited growth result in lower *L. monocytogenes* levels (up to 1.4 log) after 34 days of storage compared with the levels inoculated on the fish prior to treatments.

### 3.4. Total Viable Counts, Microbiota and pH in CS Salmon Produced with and without Preservative Fermentate P-NDV and Treated with Nisin and PGL Phages and Nisin

Total viable counts (TVCs) of mildly smoked salmon (from Experiment 2) produced with 3% NaCl and stored for 4 °C increased from 2.0 log cfu/g at the beginning of the experiment to 6.9 log cfu/g after 7 days and remained fairly stable with 6.5 log cfu/g after 29 days of storage. Counts in standard CS salmon treated with PGL phages and nisin (6.7 log ± 0.2) did not differ from those of untreated samples (*p* = 0.251). There were also no differences (*p* = 0.108) in TVCs after 29 days of storage in treated standard and sodium-reduced CS salmon. In PGL- and nisin-treated CS salmon produced with the preservative fermentate P-NDV, total counts differed (*p* = 0.013) between standard (2.4 log cfu/g) and sodium-reduced samples (5.6 log cfu/g). A tentative hypothesis for this difference could be uneven distribution of P-NDV or other inhibiting compounds (e.g., smoke components) in the CS salmon resulting in variations in bacterial growth between the samples.

The microbiota in both non-treated and treated (PGL and nisin) standard and sodium-reduced CS salmon was dominated by *Photobacterium* spp. after storage for 29 days at 4 °C. A shift in the microbiota with the dominance of *Lactococcus* spp. and *Vagococcus* spp. was observed in 29-day-stored sodium-reduced samples containing P-NDV (Figure 4). Microbiota analysis on standard CS salmon containing P-NDV was excluded due to the low levels in TVC (see above). Neither partial sodium replacement using KCl nor treatments with PGL and nisin provided changes in the pH of the CS salmon (pH 6.12 ± 0.04, 29 days storage at 4 °C).

## 4. Discussion

Ready-to-eat smoked fish products including CS salmon are foods that are easily contaminated and support the growth of pathogenic *L. monocytogenes* through their shelf-life of 4–6 weeks. They are therefore ranked first among risk food products that most often harbor *L. monocytogenes* and in levels exceeding the critical limit of 100 cfu/g at the end of shelf-life [17]. Another health consideration of CS salmon is their levels of sodium salts [23,24,51] that may exceed the recommended salt target of 3% NaCl in such products. Excess dietary sodium intake increases the risk of development of serious health complications [52,53,54]. Producing microbiologically safe and healthy CS salmon, ensuring the levels remain below critical limits of *L. monocytogenes* and with sodium levels complying with the 3% NaCl target will be major achievements for the CS salmon industry.

The lack of critical control points in the conventional production of CS salmon cannot ensure listeria-free products. A critical limit of 100 cfu/g of *L. monocytogenes* at the end of shelf-life exists in many countries (including EU member states), while others (e.g., USA) practice a zero tolerance for *L. monocytogenes* in CS salmon. Therefore, both listericidal and growth-inhibiting, bio-based prevention strategies applied directly to the product are of increasing relevance. In the present study, commercialized preparations of listeria P100 phages (PGL) and nisin did reduce *L. monocytogenes* levels on CS salmon, but listeria surviving the treatments were able to grow during storage under certain conditions. Regrowth of surviving bacteria was particularly evident at an abusive storage temperature (8 °C) or for salmon without preservative fermentates. For phage treatments, regrowth has been hypothesized to be caused by a reduction in phage titer over time [55], the emergence of phage-resistant strains [56], inhibition of phage absorption affecting the ability of the phages to infect and kill the target bacteria [57,58,59]. The observed initial reductions in listeria followed by regrowth are in line with the bactericidal action of phages upon application, but with no further technical effects during the shelf-life of the product. Consequently, the use of anti-listerial phages in foods could be regarded as processing aids. The stability of phages applied to food depends on the parameters of the food product and on the environment where it is applied [60,61]. CS salmon is a challenging product for phage treatment due to the surface microstructure that influences phage diffusion, limits access to the bacteria by the phages and thereby, restricts effective bacteria–phage interactions [62].

Phages and nisin applied separately demonstrated overall significant listericidal effects, with additive effects obtained when applied in combination. The improved listericidal effects when PGL and nisin worked together are in line with Soni et al. [43]. When applied to CS salmon containing the growth-inhibiting acetate-based preservative salts (fermentates) Verdad N6 or P-NDV, both the initial kill and subsequent growth inhibition resulted in, in most cases, 1-2 log lower *L. monocytogenes* levels in treated compared to untreated CS salmon at the end of storage. This is of considerable value since contamination levels of CS salmon are usually low [17,63]. These treatments would therefore ensure the majority of *L. monocytogenes*-contaminated CS salmon products remain below the critical limit of 100 cfu/g during the entire shelf-life.

We used high contamination levels (5 × 10^4^ cfu/cm^2^) to be able to easily detect and quantify listeria reductions. The high inoculum enabled *L. monocytogenes* under certain conditions to reach levels close to their maximum (approximately 7–8 log cfu/g), where reduced growth was observed. This appeared to reduce the difference in *L. monocytogenes* levels in treated or non-treated samples. The listeria-reducing effects may then diminish over time since the growth in treated samples eventually catches up with the levels in untreated samples. The reason for reduced growth is unknown but may be caused by competition from other bacteria, microbial metabolites (e.g., lactic acid) produced during storage or lack of micronutrients.

Growth inhibition varied with processing and storage parameters. Variations in parameters, e.g., batches of fresh salmon, dry salting (sodium-based vs. sodium-replaced mineral salts), types and levels of fermentates, smoking procedures and storage temperatures, were evident in Experiments 1 and 2 using research pilot plant-processed CS salmon and Experiment 3 using industrial smokehouse-processed salmon. As biological items, one must expect some influence from natural variations in the raw salmon (e.g., microbial and nutritional status, content of fat, pH). The importance of storing at a low, recommended temperature (4 °C) vs. abuse (8 °C) to limit the growth of *L. monocytogenes* was clearly observed in this study as in previous studies [25,44]. *L. monocytogenes* grow more rapidly and to higher levels in mildly smoked vs. more extensively smoked salmon [44]. Smoking parameters are not standardized in CS salmon production. The growth differences observed in the CS salmon in this study are likely primarily due to variations in the degree of smoking of the products (milder smoking in Experiment 2; Table 1). Other likely variations include background microbiota (type and levels of microorganisms) on the raw and processed salmon. Uneven distribution of salt may also affect *L. monocytogenes* growth in the samples. Despite these variations, overall similar effects of the various listeria-controlling treatments were obtained in the three separate experimental setups. The results therefore support that the listeria-controlling strategies evaluated in the current study will exert similar effects in most variants of industrially produced CS salmon.

Exchange of NaCl with other mineral salts does not change the ability of *L. monocytogenes* to grow in CS salmon [25]. Importantly, the present study also showed no systematic difference in the anti-listerial effects of nisin, listeria phages and preservative salts in standard CS salmon produced with NaCl and in sodium-reduced products. We did not observe different listericidal effects of PGL or nisin in CS salmon, with or without the acetic-acid-based preservative fermentates. Thus, the effects of these mitigation strategies on CS salmon appeared to be independent of the use of NaCl or mineral salt replacers and the presence or absence of preservative fermentates.

Differences in susceptibility of listeria towards commercially available phage preparations (e.g., PGL) exist, including isolates being intrinsically resistant [57,59]. Likewise, susceptibility to nisin appears to be strain-dependent [64,65]. The use of multiple strains, preferably having different phenotypic and genotypic properties, is in line with common recommendations of the EURL Lm Technical Guidance Document on challenge tests and durability studies for assessing shelf-life of ready-to-eat foods related to *L. monocytogenes* [66]. The overall anti-listerial effects of treatments with PGL were significant, although slightly reduced, in treated CS salmon contaminated with the 10-strain mix (containing two strains with reduced susceptibility to PGL) compared to salmon contaminated with the 8-strain mix (all being susceptible to PGL). The results are in line with reduced phage-mediated killing of strains with reduced phage susceptibility [57]. Surviving *L. monocytogenes* that were not killed by the phage treatment could then continue to grow, especially in the absence of growth-inhibiting fermentate or at an abusive storage temperature. However, the strain mix used (8-strain or 10-strain) explained less than 2% of the variations in the Experiment 1 dataset during storage, while fermentate Verdad N6 and the treatments explained ≥85% of these variations at both 4 °C and 8 °C storage (ANOVA Appendix A). All strains of the 10-strain mix in the current study were susceptible to nisin in a liquid assay [67], and the similar reductions in *L. monocytogenes* on nisin-treated salmon in Experiments 1 through 3 are in line with this.

The most effective control of *L. monocytogenes* was obtained using combined treatments of PGL and nisin on CS salmon containing growth-inhibiting preservative salts (Verdad N6 or P-NDV). Increased overall effects of phage and nisin treatments could have been obtained using other experimental parameters. Lower contamination levels while maintaining a high phage titer enhanced the listericidal effects of the phage treatments [57,61] and could have provided reduced regrowth of surviving bacteria during storage. For nisin, reductions in *L. monocytogenes* on treated CS salmon were dependent on nisin levels [64]. However, the selection of the most appropriate anti-listerial strategies including the type, levels and combination of hurdles is also a cost–benefit issue for the food industry [39]. One factor to keep in mind is that the use of several hurdles, e.g., combined treatments using phages, nisin and preservative salts/fermentates, will also reduce the probability of emergence and selection of mutants and tolerant strains. These are concerns that have been raised regarding the routine application of phage and nisin treatments in foods [62,68]. Of relevance and major interest is also the finding that exposure of *L. monocytogenes* serovar 4b to anti-listerial bacteriophages led to mutations in cell wall teichoic acid glycosylation genes that provided mutants that were phage-resistant but also had attenuated virulence properties caused due to their reduced cellular invasiveness [69].

Total viable counts were similar in non-treated (control) and treated (PGL and nisin) standard and sodium-reduced CS salmon after 29 days of storage. The treatments did not affect the overall microbiota on the CS salmon, which was dominated by *Photobacterium* spp. Lower bacterial numbers were present in CS salmon with the preservative salt P-NDV. The acetate-based fermentate P-NDV also showed a shift in the CS salmon microbiota with the dominance of more acid-tolerant Gram-positive bacterial genera (*Lactococcus* and *Vagococcus*) and reduced levels of *Photobacterium*. The data confirm similar effects of acetate-based fermentates on the background microbiota in sodium-replaced CS salmon (this study) as in standard CS salmon [44]. Of interest, *Photobacterium* is a potential spoiler of salmon [70], while *Vagococcus fluvialis* was identified to cause reduced levels of off-odors and thereby provide increased sensory quality in another salmon product (gravlax) with long shelf-life [71]. In summary, the microbiota of CS salmon depends on a number of factors. The role of specific bacterial species and their interaction with key processing parameters to ensure superior quality CS salmon products requires further studies.

Treatments with phages and nisin applied at exceedingly small amounts do not change the quality and sensory perception of foods [72,73]. Sensory analyses were therefore not performed on PGL- and nisin-treated CS salmon. Our recently reported full descriptive sensory analysis performed on the same batch of industrially produced standard and sodium-reduced CS salmon with and without the preservative fermentate P-NDV, as applied in Experiment 3 of the current study, also showed small differences in 23 evaluated sensory properties among standard and sodium-reduced (30% substitution of NaCl with KCl) CS salmon produced with and without P-NDV [25]. Other studies have indicated replacement of 20–40% of NaCl by KCl has little effect on the sensory properties of salmon or trout [23,74,75].

In conclusion, the present study shows that commercialized listeria-specific phages (PGL) and nisin provide significant and similar reductions in *L. monocytogenes* on contaminated standard and sodium-reduced CS salmon. Growth of *L. monocytogenes* surviving the treatments was dependent on storage temperature, but the inclusion of acetate-based preservative salts could provide complete growth inhibition during 4–5 weeks of storage. Evaluation of standard and sodium-reduced CS salmon produced in an industrial smokehouse verified that these bio-based preservation strategies provided robust anti-listerial effects on salmon processed and stored under various relevant conditions. The current study along with our recent one [25] is appropriate in supporting decision-making for the production of safe, healthy and high-quality CS salmon with reduced listeria risks, reduced sodium content and excellent sensory quality.

## Figures and Tables

**Figure 1 foods-12-04391-f001:**
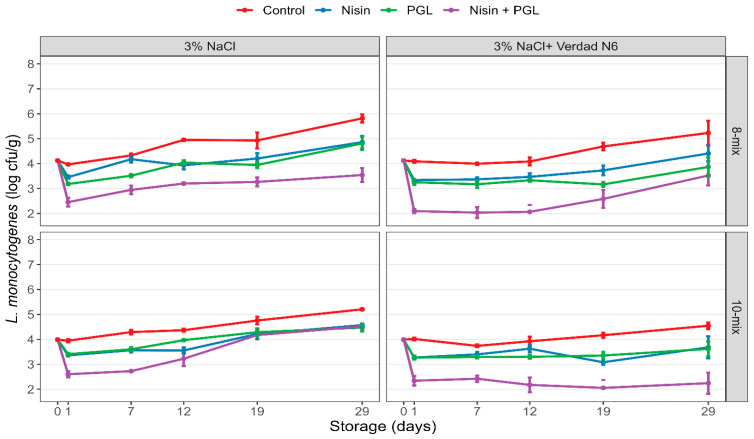
Reductions and growth of *L. monocytogenes* in cold-smoked salmon during 29 days of storage at 4 °C. Slices of CS salmon of two types (produced with 3% NaCl and 3% NaCl + 1% Verdad N6) were inoculated with two strain mixes (8-strain mix and 10-strain mix; Table 2). The contaminated samples were treated with nisin (1 ppm), listeria phages (PGL; 5 × 10^7^ pfu/cm^2^) or both (nisin + PGL) or left untreated (Control) prior to vacuum-packaging and storage. Bacterial levels were determined prior to treatments (day 0) and during storage (days 1, 7, 12, 19 and 29). Standard errors are shown.

**Figure 2 foods-12-04391-f002:**
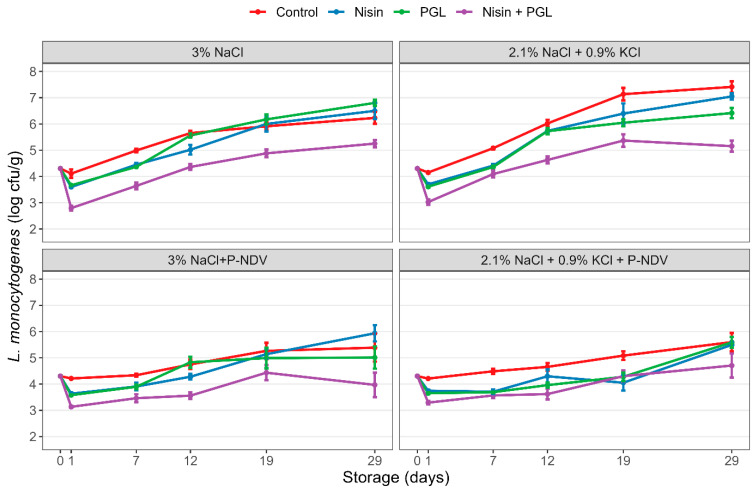
Reductions and growth of *L. monocytogenes* during 29 days of storage at 4 °C in standard (3% NaCl) and sodium-reduced (2.1% NaCl + 0.9% KCl) mildly cold-smoked and sliced salmon produced without or with the fermentate P-NDV (0.9%). The contaminated samples were treated with nisin (1 ppm), listeria phages (PGL; 5 × 10^7^ pfu/cm^2^) or both (nisin + PGL) or left untreated (Control) prior to vacuum-packaging and storage. Bacterial levels were determined prior to treatments (day 0) and during storage (days 1, 7, 12, 19 and 29). Standard errors are shown.

**Figure 3 foods-12-04391-f003:**
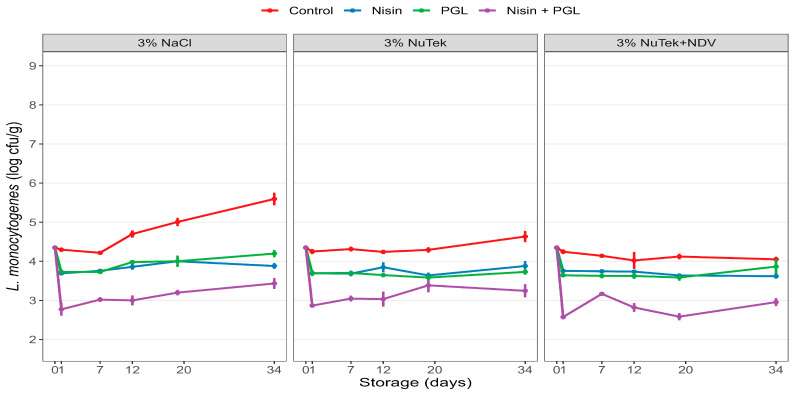
Reduction and growth of *L. monocytogenes* (10-mix) in CS salmon made by a commercial CS salmon producer. The CS salmon was produced with NaCl and sodium-reduced salt using NuTek and combined NuTek and fermentate P-NDV. The CS salmon was contaminated with a 10-strain mix of *L. monocytogenes* (Table 2) and thereafter treated with nisin (1 ppm), listeria phages (PGL, 5 × 10^7^ pfu/cm^2^) or both or left untreated (Control) and stored vacuum-packed at 4 °C for 34 days. Bacterial levels were determined prior to treatments (day 0) and during storage (days 1, 7, 12, 19 and 34). Standard errors are shown.

**Figure 4 foods-12-04391-f004:**
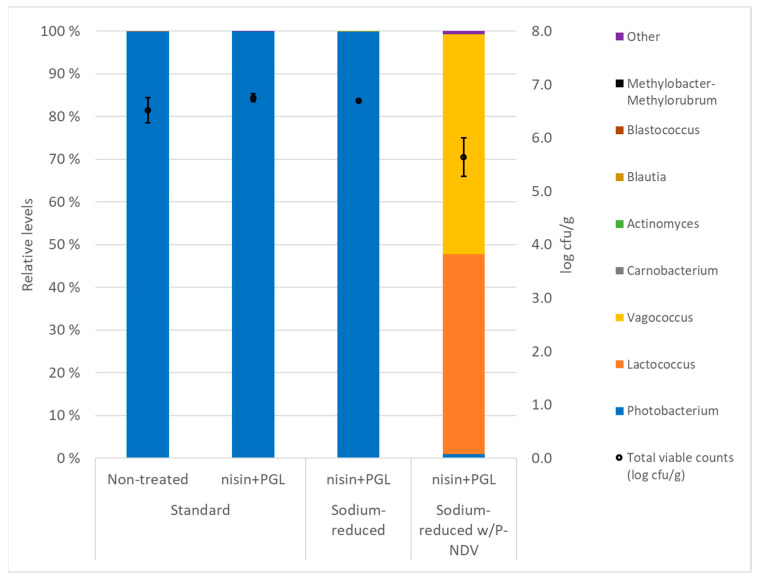
Total viable counts and microbiota after storage of vacuum-packed, mildly CS salmon at 4 °C for 29 days. The CS salmon were of three types: standard (produced with 3% NaCl), sodium-reduced (produced with partial replacement of NaCl with KCl) and sodium-reduced with added fermentate P-NDV in the salting process. The CS salmon was either untreated or treated with nisin (1 ppm) and listeria phages (PGL, 5 × 10^7^ pfu/cm^2^) before vacuum-packaging and storage. Dominant bacterial groups were represented by different colors. The taxa are colored according to family or genus affiliation. The data for each type of CS salmon and treatment are mean values of three samples (data are available for sodium-reduced CS salmon type for only one sample). Only taxa with average over all samples above 0.2% or value above 2% are presented. The remaining taxa are represented as “Other”.

**Table 1 foods-12-04391-t001:** Variables and parameters for the three separate anti-listerial treatment and storage experiments (Experiment 1, 2, 3) ^1^.

Variables	Experiment 1	Experiment 2	Experiment 3
Site of production	Pilot plant	Pilot plant	Industrial smokehouse
Standard CS salmon	3% NaCl	3% NaCl	3% NaCl + 0.6% sucrose
Sodium-reduced CS salmon	Not included	2.1% NaCl + 0.9% KCl	3% NuTek 78300
Preservative salt (fermentate)	Without or with Verdad N6	Without or with P-NDV	Without or with P-NDV in sodium-reduced CS salmon
Smoking	Moderate	Mild	Moderate
*L. monocytogenes* mix ^2^	8-strain and 10-strain	10-strain	10-strain
Storage temperature	4 °C and 8 °C	4 °C	4 °C
Storage time	29 days	29 days	34 days

^1^ See Section 2 for further experimental details. ^2^ See Table 2 for *L. monocytogenes* strain information.

**Table 2 foods-12-04391-t002:** *L. monocytogenes* strains used in the present work ^1^.

Strain No.	Serotype	MLVA/ST ^2^	Source ^3^	Other Designations; Reference
MF3860	1/2a	6-10-5-16-6/20	Salmon processing, Plant S4	[10]
MF3939	1/2a	5-8-15-10-6/14	Salmon processing, Plant S3	[10]
MF4001	1/2a	5-8-15-10-6/14	Salmon processing, Plant S2	[10]
MF4077	1/2a	6-9-18-16-6/8	Salmon processing, Plant S1	[10]
MF4588	1/2a	7-7-10-10-6/7	Salmon processing, Plant S1	[10]
MF4804	1/2a	6-7-14-10-6/121	Salmon processing, Plant S2	[10]
MF2184	1/2b	7-8-0-16-0/3	Meat processing, outbreak	2583/92; [48]
MF3009	1/2b	n.d./5	Cattle	FSL J2-064; [49] https://www.ncbi.nlm.nih.gov/nuccore/AARO00000000.2/ (accessed on 3 October 2023)
MF3039	4b	n.d./6	Human, cerebrospinal fluid, outbreak	FSL N1-227; [50] https://www.ncbi.nlm.nih.gov/pmc/articles/PMC3889766/ (accessed on 3 October 2023)
MF3710	4b	7-7-20-6-10/n.d.	Human, cerebrospinal fluid	CCUG3998; Culture Collection University of Gothenburg

^1^ For the 8-mix strain, strains MF3860 and MF3039 were omitted due to lower sensitivity to the listeria phages. ^2^ MLVA designation according to Møretrø et al. [10]. ST numbers refer to the Institute Pasteur MLST database (https://bigsdb.pasteur.fr/, accessed on 23 April 2022). n.d.: not determined ^3^ Plant designation according to Møretrø et al. [10].

**Table 3 foods-12-04391-t003:** Anti-listerial effects of CS salmon treated with nisin, PGL or nisin + PGL during the storage period for 29 days at 4 °C.

Experiment No. ^1^	Treatment ^2^	Reductions (log) in *L. monocytogenes* Levels during Storage ^3^				
		Day 1	Day 7	Day 12	Day 19	Day 29
Exp. 1	Nisin	0.7 (**) ^4^	0.5 (.)	0.7 (**)	0.8 (***)	0.8 (***)
	PGL	0.7 (**)	0.7 (**)	0.7 (**)	1.0 (***)	1.0 (***)
	Nisin + PGL	1.6 (***)	1.6 (***)	1.7 (***)	1.6 (***)	1.7 (***)
Exp. 2	Nisin	0.5 (*)	0.6 (**)	0.4 (*)	0.5 (*)	−0.1 (ns)
	PGL	0.6 (**)	0.6 (***)	0.3 (ns)	0.5 (*)	0.2 (ns)
	Nisin + PGL	1.1 (***)	1.0 (***)	1.2 (***)	1.1 (***)	1.4 (***)

^1^ See Section 2 and Table 1 for details of Experiments 1 and 2. ^2^ The CS salmon was treated with nisin (1 ppm), listeria phages (PGL, 5 × 10^7^ pfu/cm^2^) or both. ^3^ The numbers represent the reductions (log) in *L. monocytogenes* levels obtained by the treatments compared with non-treated CS salmon. They were determined as the average reductions in the means per day and type of CS salmon (standard (3% NaCl) and sodium-reduced (2.1% NaCl + 0.9% KCl) with and without the fermentates Verdad N6 (Exp. 1) and P-NDV (Exp. 2). Experiment 1: averaged data of the 8- and 10-strain mix used are reported; Experiment 2: data on the 10-strain mix used are reported. ^4^ Significance levels: ns = nonsignificant (*p* > 0.1); *. p* = 0.05–0.1; * *p* = 0.01–0.05; ** *p* = 0.001–0.01; *** *p* ≤ 0.001.

**Table 4 foods-12-04391-t004:** Anti-listerial effects of nisin, PGL or nisin + PGL on industrially processed standard and sodium-reduced CS salmon during storage for 34 days at 4 °C.

Treatment ^1,2^	Overall Reductions (log) in *L. monocytogenes* in Three Types of CS Salmon during Storage ^3^		
	Standard (3% NaCl)	Sodium-Reduced (3% NuTek)	Sodium-Reduced (3% NuTek) + P-NDV
Nisin	0.9	0.6	0.4
PGL	0.8	0.7	0.5
Nisin + PGL	1.7	1.2	1.3

^1^ See Section 2.4 and Table 1 for details of Experiment 3. ^2^ The CS salmon was treated with nisin (1 ppm), listeria phages (PGL, 5 × 10^7^ pfu/cm^2^) or both. ^3^ The numbers represent the overall reductions (log) in *L. monocytogenes* levels obtained by the treatments compared with non-treated CS salmon. The reductions were determined as the average reductions in the means over all days (sampling days 1, 7, 12, 19 and 34). All reductions were significant at level *p* ≤ 0.001.

## Data Availability

Data is contained within the article or Appendix A.

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
