# Peer review of "Reduction and Growth Inhibition of Listeria monocytogenes by Use of Anti-Listerial Nisin, P100 Phages and Buffered Dry Vinegar Fermentates in Standard and Sodium-Reduced Cold-Smoked Salmon"

_foods, 2023, doi:10.3390/foods12244391_

Round 1

Reviewer 1 Report

Comments and Suggestions for Authors

The authors presented extensive work on the control of listeria growth in cold-smoked salmon. The research topic is relevant and interesting to readers.

The authors described the results of 3 experiments. This complicates the general perception and detailed analysis of the results obtained, but shows the results of the study in a comprehensive manner. The research results are presented clearly; the number of figures and tables is sufficient.

In this study, the authors emphasized microbiological studies. The authors selected factors that correspond to the conditions of industrial production and further storage of salmon. The authors used salts and commercial fermentates to process salmon. It would be useful for readers to know the composition of commercial fermentates (lactate and/or acetate content). To assess the anti-listerial barrier, it is also important to know the amount of salts (chlorides, lactates, acetates) in the finished product, the amount of moisture, pH and water activity (in the discussion, the authors noted that the processing of salmon may have been uneven). These indicators may be important in assessing the reduction in microbial growth achieved.

Section 2.2. Describe in more detail the procedure for obtaining a suspension of L.monocytogenes (10 4 CFU/ml) for salmon contamination.

Why were suspensions of 8 and 10 strains chosen for infection? What was this connected with?

Section 2.5. Please specify the purpose for which two 5g pieces of salmon (cooked and uncooked) were combined to produce a 10g sample.

The title of the article should be shortened as the results did not show inhibition of listeria growth, only reduction.

Author Response

Author’s reply to the Review Report (Reviewer 1)

The reviewers suggestions and comments are given with author’s responses in blue:

The authors presented extensive work on the control of listeria growth in cold-smoked salmon. The research topic is relevant and interesting to readers.

The authors described the results of 3 experiments. This complicates the general perception and detailed analysis of the results obtained, but shows the results of the study in a comprehensive manner. The research results are presented clearly; the number of figures and tables is sufficient.

Thank you for the positive response!

In this study, the authors emphasized microbiological studies. The authors selected factors that correspond to the conditions of industrial production and further storage of salmon. The authors used salts and commercial fermentates to process salmon. It would be useful for readers to know the composition of commercial fermentates (lactate and/or acetate content). To assess the anti-listerial barrier, it is also important to know the amount of salts (chlorides, lactates, acetates) in the finished product, the amount of moisture, pH and water activity (in the discussion, the authors noted that the processing of salmon may have been uneven). These indicators may be important in assessing the reduction in microbial growth achieved.

Response: The composition of the commercial fermentates is not known.The product data sheets of the fermentates applied in the current study, Verdad N6 and Provian NDV, provide no information on the content of lactates and acetates in the fermentates. However, relevant information of the fermentates is described in the manuscript (Section 2.3).

We agree that the mentioned parameters (levels of salts, lactates/acetates, pH and water activity) as well as others (e.g. phenol content according to degree of smoking of the cold smoked (CS) salmon and background microbiota differences) could influence on the anti-listerial barrier in this type of products. Cold smoked salmon is not a defined product and levels of salt, organic acid salts, pH, aw and smoking compounds may vary. The main intention of this study was to evaluate the effects of anti-listerial treatments on salmon with variations in certain relevant key processing, product and storage parameters and thereby obtain valuable information on the ability of these strategies to provide robust anti-listerial effects on CS salmon produced and stored under relevant although variable conditions. Evaluations of the effects of some specific parameters that could influence on L. monocytogenes growth and survival of the products were therefore not a main objective. However, pH was measured and reported (section 3.4) with no changes in pH recorded according to sodium replacement nor treatment with PhageGuard L and nisin. Our recent study (Heir et al. 2022; Microbial Safety and Sensory Analyses of Cold-Smoked Salmon Produced with Sodium-Reduced Mineral Salts and Organic Acid Salts-Web of Science Core Collection) also showed no effects on pH and aw in standard and sodium-substituted products and that >90% of the variation in L. monocytogenes levels during storage of such CS salmon products was explained by storage time and temperature, both parameters included in the current study. In CS salmon processing, some variations in product parameters within and between CS salmon production batches and fillets must be expected as mentioned in the Discussion. However, the current study showed that similar effects of the various anti-listerial treatments were obtain through the three experimental setups implying that robust effects could be obtained despite certain variations in process and product parameters of CS salmon. This was a key finding in the assessment of the suitability of the tested treatments in the current study. No changes in the revised manuscript.

Section 2.2. Describe in more detail the procedure for obtaining a suspension of L.monocytogenes (10 4 CFU/ml) for salmon contamination.

Response: The procedure in section 2.2 has been described in more detail as suggested. Further details have also been provided in section 2.5 to provide a clear and correct description of the procedure for contaminating the salmon with the L. monocytogenes suspensions from section 2.2.

Why were suspensions of 8 and 10 strains chosen for infection? What was this connected with?

Response: The use of a mix of L. monocytogenes strains is in line with recommendations for performing challenge tests and durability studies on ready-to-eat foods relate to L. monocytogenes. We have therefore in this and in previous studies (e.g. Heir et al. 2022 (see above); Heir et al. 2019 Reduction and inhibition of Listeria monocytogenes in cold-smoked salmon by Verdad N6, a buffered vinegar fermentate, and UV-C treatments-Web of Science Core Collection) used a mix of 10 strains including strains of common serovars and sequence types from salmon processing as well as isolates from listeriosis outbreaks. We therefore wanted to use the same mix of strains in this study. However, two of the strains in the 10-strain mix showed reduced susceptibility towards PhageGuard L (PGL) (as reported in the Results (section 3.1) and in the Discussion). It was therefore of interest to compare the effects of the anti-listerial treatments on strain mixes containing strains with different susceptibility towards PGL. The background for using multiple strains and results using 8- or 10 strain mix of L. monocytogenes are discussed in a separate paragraph in the Discussion (revised MS line 542-558).

Section 2.5. Please specify the purpose for which two 5g pieces of salmon (cooked and uncooked) were combined to produce a 10g sample.

Response: The addition of a second sliced piece of CS salmon on the top of the contaminated and treated first slice of salmon followed by packaging was to reflect a scenario where the salmon was contaminated and treated during slicing and then packed and stored as a sliced product. Additional brief information has been included in section 2.5.

The title of the article should be shortened as the results did not show inhibition of listeria growth, only reduction.

Response: The study showed that the presence of fermentates (Verdad N6 and Provian NDV) added as ingredients in the salting process of CS salmon production provided reduced growth of contaminating L. monocytogenes during storage. Even complete growth inhibition can be obtained using fermentates as indicated in Figure 1. Although complete growth inhibition cannot be assured, the results show that increased lag time and reduced growth rate leading to lower levels of L. monocytogenes in contaminated products during storage is a likely scenario when fermentates are applied in the production of CS salmon. These growth inhibitory effects are likely to reduce food safety risks and outcome of listeria contaminated, stored CS salmon products. We therefore suggest keeping the original title.

Reviewer 2 Report

Comments and Suggestions for Authors

The topic is interesting. Reducing the food safety risk of cold-smoked salmon caused by Listeria monocytogenes is an important topic. In addition to investigating alternative, bio-based Listeria inhibition methods, the authors also investigated the possibility of the reduction of NaCl. Using bacteriocin and phage is a promising and very interesting method.

The manuscript is well written.

The methods chosen are appropriate. The presentation of the results is adequate. The conclusions are good.

However, minor clarifications are needed:

Line 60: It would be better to number the link as a reference and include it in the reference list.

Line 70: The abbreviation „RTE” is not explained.

Line 137: The abbreviation „MLVA” is not explained.

Line 512-516: „Neither did we observe different listericidal effects of PGL or nisin in CS salmon with or without the acetic acid based preservative fermentates. Thus, the effects of these mitigation strategies  on CS salmon appeared to be independent of the use of NaCl or mineral salt replacers and the presence or absence of preservative fermentates.”

This statement is not entirely clear. In the Experiment 1 in the case of the Nisin + PGL treatment, the sample treated with Verdad N6 showed a much larger log reduction (Fig 1), and the ANOVA analysis (Table S2, S3) also shows that the Verdad has a great proportion in the explained variance and the authors wrote in line 295-296 „ANOVA showed treatments and Verdad N6 to explain the vast majority (≥85 %) of the variability in the data”.

Line 533: Instead of „Table xxx” is Table S2, S3.

If the shelf life is 4-6 weeks, why were the samples not tested for 6 weeks? Why were there different storage times in the Experiment 1, 2 and Experiment 3?

Table 3.:

Does the average log number provide adequate information in this form? If averaging the data of samples treated with or without Verdad N6 in the Experiment 1 and with salt or reduced salt, with or without P-NDV in the Experiment 2, does not the effect of these parameters show up in the averages and give an inappropriate conclusion about the results of nisin, PGL, nisin + PGL treatments? Do these averages relate to the 8-strain or 10-strain mix?

Author Response

Author’s reply to the Review Report (Reviewer 2)

The reviewer suggestions and comments are given with author’s responses indicated in blue:

The topic is interesting. Reducing the food safety risk of cold-smoked salmon caused by Listeria monocytogenes is an important topic. In addition to investigating alternative, bio-based Listeria inhibition methods, the authors also investigated the possibility of the reduction of NaCl. Using bacteriocin and phage is a promising and very interesting method.

The manuscript is well written.

The methods chosen are appropriate. The presentation of the results is adequate. The conclusions are good.

Response: Really appreciate your positive response to our manuscript!

However, minor clarifications are needed:

Line 60: It would be better to number the link as a reference and include it in the reference list.

Response: Done. The reference is reference no. 21 in the reference list

Line 70: The abbreviation „RTE” is not explained.

Response: Explanation included

Line 137: The abbreviation „MLVA” is not explained.

Response: Corrected.

Line 512-516: „Neither did we observe different listericidal effects of PGL or nisin in CS salmon with or without the acetic acid based preservative fermentates. Thus, the effects of these mitigation strategies on CS salmon appeared to be independent of the use of NaCl or mineral salt replacers and the presence or absence of preservative fermentates.”

This statement is not entirely clear. In the Experiment 1 in the case of the Nisin + PGL treatment, the sample treated with Verdad N6 showed a much larger log reduction (Fig 1), and the ANOVA analysis (Table S2, S3) also shows that the Verdad has a great proportion in the explained variance and the authors wrote in line 295-296 „ANOVA showed treatments and Verdad N6 to explain the vast majority (≥85 %) of the variability in the data”.

Response: For Nisin + PGL treated samples, the L. monocytogenes log reductions (day 1) appear to be slightly higher in samples with Verdad N6 compared to samples without. The log difference was in the range 0.3 to 0.5 log with the highest difference observed in samples contaminated with the 8-strain mix. This is the case for Experiment 1 and Figure 1 as mentioned by the reviewer. However, this appear not to be a general response. In Figure 2, the results of Nisin + PGL treated salmon indicate higher initial reductions (day 1) in CS salmon without fermentate than in samples with fermentate. The same appears in Figure 3. Thus, opposite of the Experiment 1 data. The different initial reductions obtained in samples with and without fermentates are in all cases small. We therefore regard that the above sentence (line 512-516) is valid especially in context with the sentence proceeding this statement (line 533-535 in the revised manuscript):

“Importantly, the present study also showed no systematic difference in the anti-listerial effects of nisin, listeria phages and preservative salts in standard CS salmon produced with NaCl and in sodium-reduced products”. The ANOVA analyses are for the whole storage period showing that Verdad N6 has a significant effect of listeria levels throughout the storage period that together with the treatments explain approx. 85% of the variability in the data.

Line 533: Instead of „Table xxx” is Table S2, S3.

Response: Corrected

If the shelf life is 4-6 weeks, why were the samples not tested for 6 weeks? Why were there different storage times in the Experiment 1, 2 and Experiment 3?

Response: For Experiments 1 and 2, the storage times were identical and the same as in our previous studies on CS salmon (e.g. Heir et al. 2022 Microbial Safety and Sensory Analyses of Cold-Smoked Salmon Produced with Sodium-Reduced Mineral Salts and Organic Acid Salts-Web of Science Core Collection ; Heir et al. 2019 Reduction and inhibition of Listeria monocytogenes in cold-smoked salmon by Verdad N6, a buffered vinegar fermentate, and UV-C treatments-Web of Science Core Collection). This included storage of vacuum-packed CS salmon at 4 and 8 °C with sampling performed at day 0, 1, 7, 12, 19 and 29. For Experiment 3, the storage times were as in Experiments 1 and 2 except for the last sampling performed after 34 days storage. According to the generally limited growth of L. monocytogenes in the CS containing the fermentate Provian NDV in this experiment observed during storage, we determined to extend the storage to day 34 to sort out whether extended storage could provide increased L.monocytogenes levels in this CS salmon that were made by a commercial CS salmon producer. Extending the storage times from 29 days/ 34 days to 6 weeks as questioned by the reviewer are likely to not lead to other conclusions as the growth of L. monocytogenes are generally higher in the earlier phase of storage with limited extra growth of L. monocytogenes later in the storage period.

Table 3.:

Does the average log number provide adequate information in this form? If averaging the data of samples treated with or without Verdad N6 in the Experiment 1 and with salt or reduced salt, with or without P-NDV in the Experiment 2, does not the effect of these parameters show up in the averages and give an inappropriate conclusion about the results of nisin, PGL, nisin + PGL treatments?

Do these averages relate to the 8-strain or 10-strain mix?

Response:

In all cases we have compared the effect of the treatments (nisin, PGL, nisin + PGL) in similar CS salmon e.g. the effect of nisin in Verdad-treated salmon was compared to L.m. levels in non-treated Verdad salmon to determine the effects nisin. Therefore, the effects of Verdad do not interfere with the reported effects of the nisin treatment. Since the effects of e.g. fermentates do not interfere, it is appropriate to average the results. The implications are that the treatments are expected to have an effect for all the types included in these experiments demonstrating the robustness of the treatment in different types of CS salmon. We therefore regard the data in Table 3 are adequate as the table shows an overall effect of the treatments across different types of smoked salmon demonstrating the robustness of the treatments.

In Experiment 1 the effect of mix (10-strain and 8-strain) is averaged out, whereas in Experiment 2 this is for 10-mix according to info in Table 1. Results comparing 8- and 10-mix in experiment 1 are described in the text, lines 297-301. Information on strain mixes used in Experiment 1 and 2 has been included in a footnote of Table 3.